# Amendment of Livestock Manure with Natural Zeolite-Clinoptilolite and Its Effect on Decomposition Processes during Composting

Eva Šubová [1], Naďa Sasáková [1,*], František Zigo [2] , Ingrid Mindžáková [1], Mária Vargová [1], Ján Kachnič [1] and Katarína Veselitz Laktičová [1]

[1] Department of Public Veterinary Medicine and Animal Welfare, University of Veterinary Medicine and Pharmacy, 041 81 Košice, Slovakia; eva.subova@uvlf.sk (E.Š.); ingrid.mindzakova@uvlf.sk (I.M.); maria.vargova@uvlf.sk (M.V.); jan.kachnic@uvlf.sk (J.K.); katarina.veszelitslakticova@uvlf.sk (K.V.L.)

[2] Department of Nutrition and Animal Husbandry, University of Veterinary Medicine and Pharmacy, 041 81 Košice, Slovakia; frantisek.zigo@uvlf.sk

[*] Correspondence: nada.sasakova@uvlf.sk; Tel.: +421-915-984-672

**Abstract:** The aim of study was to investigate the effect of amendment of cow manure with natural zeolite-clinoptilolite and hydrated lime on decomposition processes over the period of 90 days. Two static piles of amended substrates were constructed consisting of cow manure with an addition of bulking material (2.5% by weight): (1) manure mixed with zeolite (S1); manure mixed with zeolite and lime (S2). Third amendment-free pile served as a control (C). During the experiment, pH level, temperature (T), dry matter (DM), ash, organic matter (OM), C/N ratio, ammonia nitrogen ($N$-$NH_4^+$) and total nitrogen ($N_t$) were determined. We also determined the counts of total coliform and faecal coliform bacteria and faecal streptococci as indicators of the hygiene level of compost. A significant increase ($p < 0.001$) in temperature to 53 °C was observed in S2 compared to C. In S2 we observed a significantly reduced release of $N$-$NH_4^+$ from the composting substrate compared to C ($p < 0.05$). The significant differences were in $N_t$ content in C and S2 ($p < 0.001$) and between S1 and S2 ($p < 0.05$). The concentration of $N_t$ increased and caused decrease in the C/N ratio. The content of $N_t$ in the substrates with zeolite increased by 44% in S1 and 45% in S2 compared to C. The differences in counts of coliform and faecal coliform bacteria between C and S2 were significant ($p < 0.001$). This experiment showed that amendment with zeolite and lime decreased nitrogen losses during composting and indicated sorption effects of zeolite.

**Keywords:** composting; zeolite; nitrogen; physicochemical factors; microorganism

## 1. Introduction

Livestock industry is producing large quantities of organic waste in the form of animal excrements with potential serious impact on the environment. Improper handling, processing, storing and application of animal manure presents hygiene and epidemiological problems [1,2]. On the other hand, organic wastes can be a source of organic matter, nitrogen, phosphorous, potassium and trace elements. Due to their high nutrient content, after relevant processing they can be used as a valuable organic fertilizer [3]. Waste management must be properly managed to prevent potential environmental pollution. Improper storage and processing of organic wastes can lead to environmental pollution, eutrophication and emission of greenhouse gases and result in loss of the economic value of wastes. Effective utilization of organic wastes requires a risk assessment to prevent adverse environmental pollution effects on the air, soil and surface water [4].

Composting is considered as one of the safest means of effective management of manure and mitigation of environmental problems caused by animal wastes [5] with many benefits. In Slovakia, composting is a common practice in cattle manure management.

When stored under controlled conditions, manure turns to a hygienic material, free of odour, viable pathogens and weed seeds [6]. Composting is a complex process during which organic substrates undergo substantial bio-transformation. During this process, by the action of various microorganisms and fungi, diverse physical and biological changes take place [7]. Therefore, the final composting product is suitable for agricultural use as a fertilizer and becomes part of a sustainable waste management strategy [8]. Onwosi et al. [9] summarized the main factors affecting composting including the temperature, C/N ratio, moisture content, electrical conductivity, aeration, pH and particle size. These factors can change continuously throughout the decomposition process and significantly affect the final product [10]. During composting, constant variation of physical and chemical parameters induces changes in the composition and activities of the present microbial communities. In the composted substrate, the composition and activity of microbial communities is affected by the constant variation of physicochemical parameters [11]. Under the action of constantly changing bacterial communities the organic portion of manure (proteins, lipids, cellulose, lignin) is degraded [12].

Study by Meng et al. [7] showed that $N-NO_3$, $N-NH_4^+$, total nitrogen, C/N ratio, temperature and moisture content significantly influenced composition of the relevant bacterial community. The most important factor controlling the reaction rate of composting is temperature that affects the microbial metabolic rate and population structure [13,14]. Temperature evolution during composting can be generally divided into three phases: mesophilic, thermophilic and mesophilic-maturation stages [15,16]. Generally, two main classes of microorganism are involved in the composting process. During the short initial period, mostly mesophilic processes are involved and are followed by the thermophilic ones. The mesophiles grow optimally at temperatures of 20–35 °C, but tolerate also range of 10–40 °C. Thermophiles are optimally adapted to temperatures in the range of 50–60 °C, although they can tolerate temperature range of 30–90 °C [9,14].

The main problem during on-farm composting is the nitrogen loss through ammonia volatilization and release of ammonia to the environment [17]. To reduce pollution and increase the value of the compost, the losses of $NH_3$, $N_2$ and $N_2O$ must be controlled [18]. Amendment of organic substrates with natural materials before composting improves utilization of nutrients, decreases leakage of pollutants, particularly of nitrogen and phosphorus into soil and water, and prevents contamination of the environment by microorganisms [3]. Additives can modify the structure of compost and optimize properties of the composted substrate. Therefore, the physicochemical and microbial changes after adding zeolite and other materials to the compost have been studied by a number of authors [18,19]. Zeolites are three-dimensional, microporous crystalline materials with well-defined structure of voids and channels that contain aluminium, silicon and oxygen in their regular framework. The most widespread and studied is the naturally occurring zeolite-clinoptilolite [20]. The main properties of zeolite are high ion-exchange capacity and high water holding capacity [21]. The cation exchange capacity of zeolite represents $100 \ mol \cdot kg^{-1}$. Zeolites exhibit high ability to adsorb ammonium and nitrate ions on its surface which inhibits its conversion to free ammonia. Thereby zeolites control nitrogen losses and reduce odours and air pollution during the composting process [22].

Gholamhoseini et al. [23] reported that the addition of 14–21% (*w/w*) of natural zeolite clinoptilolite could improve $NH_4^+$, available N, organic N and total N of a manure compost. The study by Bautista et al. [24] also showed that addition of zeolite before composting reduces ammonia emissions and improves fertilizer quality by serving as a slow-release N source. Zeolites exhibit also high thermal stability and are used as catalysts. Furthermore, they reduce emissions of greenhouse gases and ammonia during the composting process. Zeolite can upgrade the compost quality by enhancing maturity, and decreasing salinity. Addition of different doses of zeolites to the substrates before composting is recommended mainly because of differences in their composition. Overall, presence of natural zeolites in the composted substrates is beneficial [25]. Zeolite additives not only improve physicochemical properties of the product, but also enhance microbial activities and reduce the

duration of the composting process. For the sake of sustainable agriculture, production and application of zeolite-containing cattle manure compost is advantageous as it helps to reduce the application of chemical fertilizers [23].

The aim of this study was to investigate the effects of co-amendment of dairy manure with lime and zeolite during composting with respect to reduction of $N-NH_4$ and N losses. The following hypotheses were tested: (1) $NH_3$ volatilization and thus N losses may be reduced by co-amendments of lime and zeolite compared to the control; (2) amendment with zeolite alone results in increased $NH_4$ sorption by zeolite beneficial from the point of retention of nutrients and reduction of environmental pollution.

## 2. Materials and Methods

Three static composting piles were constructed in the outdoor area of the UVMP experimental facility in Košice, Slovakia. Investigations were carried out from June to August, 2019.

### 2.1. Substrates and Amendments

The piles contained the following: raw manure mixed with 2.5% by weight of zeolite (S1); raw manure mixed with 2.5% by weight of each zeolite and lime (S2); raw manure without any amendments (C) used as a control. Cow manure that consisted of dairy manure and bedding material (straw) was obtained from a cattle farm PD Paňovce, Slovakia, and was transported to the experimental area immediately before construction of the piles. The initial moisture content of the manure was 81.8%.

The natural zeolite (clinoptilolite) used in this study was obtained from Zeocem, a.s., Quarry in Nižný Hrabovec, the Slovak Republic. The natural zeolite of grey-green colour consisted principally of clinoptilolite, pH 6.8–7.2, partial exchange capacity min. $0.65$ mol·$kg^{-1}$, total exchange capacity 1.2–1.5 mol·$kg^{-1}$. Ion exchange properties of the clinoptilolite: $Ca^{2+}$ 0.64–0.98 mol·$kg^{-1}$; $Mg^{2+}$ 0.06–0.19 mol·$kg^{-1}$; $K^+$ 0.22–0.45 mol·$kg^{-1}$; $Na^+$ 0.01–0.19 mol·$kg^{-1}$. Selectivity: $Cs^+ > NH_4^+ > Pb^{2+} > K^+ > Na^+ > Ca^{2+} > Mg^{2+} > Ba^{2+} > Cu^{2+}$, $Zn^{2+}$. Chemical and mineralogical composition of the zeolite is summarized in Table 1 [26].

**Table 1.** Chemical and mineralogical composition of zeolite.

| Chemical | (%) | Mineralogical | (%) |
|---|---|---|---|
| $SiO_2$ | 64.18–75.50 | Clinoptilolite | 80–84 |
| $Al_2O_3$ | 10.93–14.80 | Cristobalite | 9 |
| CaO | 1.43–11.68 | Plagioclase | 5–8 |
| $K_2O$ | 1.24–4.24 | Clay mica | 2–3 |
| $Fe_2O_3$ | 0.12–2.45 | Quartz | traces |
| MgO | 0.29–1.43 | | |
| $Na_2O$ | 0.10–2.97 | | |
| $TiO_2$ | 0.08–0.39 | | |
| $P_2O_5$ | 0.01–0.18 | | |
| Si/Al | 4.8–5.4 | | |

Source: Central Agricultural Inspection and Testing Institute in Bratislava, 2017.

Hydrated lime is a limestone burned at temperatures of 1100 °C or higher to drive out enough $CO_2$ to leave just calcium oxide CaO. Then it is rehydrated by adding water which generates extremely alkaline calcium hydroxide Ca $(OH)_2$. This alkalinity makes the hydrated lime so useful for neutralization of acidity. Lime added to the compost pile raises the pH, making the pile more alkaline. Most compost piles have neutral or slightly alkaline pH once they have finished decomposing. The hydrated lime used in this experiment was purchased from the store (Lime hydrate CL 90-S according to ČSN EN 459-1 ISO 9001 company KOTOUČ ŠTRAMBERK, Czech Republic).



## 2.2. Construction of Piles

The area for composting was covered with a roof to protect the piles from rain. The piles were built on a concrete floor covered with boards and topped with a plastic foil. On top of the foil, there was placed a 10 cm thick layer of straw on which the mechanically homogenized substrates were piled. Air ducts allowed free access of air to the bottom of the piles. The total weight of each pile was 90 kg and the dimensions approximately 80 cm × 80 cm × 80 cm (length × width × height). Chopped straw, 4 ± 2 cm in length in the amount of 14 kg per pile was used as a bulking material. The substrates were stored for 3 months without turning in an effort to imitate the natural conditions feasible on the farms. Temperature probes were situated approximately in the core of each pile. We monitored the ambient temperature near the composting piles.

## 2.3. Analytical Determinations

The temperature levels in each pile were determined with digital thermometers Testo 175 and the results were processed by ComSoft Basic software. The changes in physicochemical properties were monitored by collection of three samples from the core of each pile after 1, 3, 5, 8, 13, 19, 26, 40, 62 and 90 days of storage. Examination of all parameters was carried out in duplicate.

Water extract (1:10) for pH determination was obtained by shaking homogenized samples for 15 min with distilled water. A pH-meter HQ440d multi of fy HACH equipped with a glass electrode was used for measurement. For determination of dry matter, samples were dried to a constant weight in an oven set to 105 °C. Ash was reported as a residuum upon incineration at 550 °C/4 h. Steam distillation and titration were used to determine ammonium ions ($NH_4^+$). For determination of total nitrogen, samples were wet-digested employing a HACH-Digesdahl apparatus and the digestate aliquot was subjected to steam distillation with 40% NaOH.

## 2.4. Bacteriological Examination

The samples used for physicochemical examination were examined also bacteriologically in the same intervals. We determined the counts of total coliforms and faecal coliform bacteria and faecal streptococci that indicated hygiene level of the substrate during composting. The respective counts were reported as means of $\log_{10}$ CFU·mL$^{-1}$ ± standard deviation. Plate counts of total coliforms and *E. coli* (CFU·mL$^{-1}$) were determined on Endo agar (HiMedia, India) with incubation for 24 h at 37 °C or 43 °C, respectively. Plate counts of faecal enterococci (CFU·mL$^{-1}$) were determined on a solid selective medium containing sodium azide and colourless 2,3,5-trifenyltetrazolium chloride.

## 2.5. Statistical Analysis

The results of physical-chemical analyses in investigated substrates (pH, DM, OM, TOC, N-$NH_4^+$, $N_t$) are presented as the means (n = 3) ± SD for all examined piles. Differences between groups were analysed by one-way ANOVA. Difference $p < 0.05$ was considered to be statistically significant.

## 3. Results

### 3.1. Temperature and pH Level

The course of temperature throughout the composting is presented in Figure 1. The ambient temperature was also monitored near the composting piles and ranged from 12.7–27.8 °C. Temperature in the core of the substrates increased during the first week due to rapid degradation of organic matter. Substrate S2 exhibited temperature pattern typical of composting: heating, thermophilic, and cooling phases and ranged from 26.3–53 °C. Furthermore, elevated temperature promoted degradation of the substrate to simpler components. In comparison with S2, lower temperatures were observed in S1 and control substrates throughout the process (25.8–38.8 °C and 23.5–41.6 °C, respectively). During the composting, the maximum temperatures reached in S1, S2 and C piles were 41.6 °C, 53 °C

and 38.8 °C, respectively. Higher temperatures (>50 °C) in the S2 pile were recorded on days 3, 4 and 5. After this period, the temperature gradually decreased to 43.3 °C which indicated the end of the thermophilic phase of composting. Between days 12 and 18, the temperature in the S2 core showed another slight increase followed by a decreasing trend resembling that in S1 pile. The temperatures exceeding 50 °C during three days in S2 substrate were probably not sufficient to eliminate potential pathogens and ensure hygiene safety of the compost. Likewise, effective pathogen removal could not be ensured in S1 and C piles. The course of temperature in the substrates investigated in our study showed a significant effect of amendment with zeolite and lime on temperatures during composting.

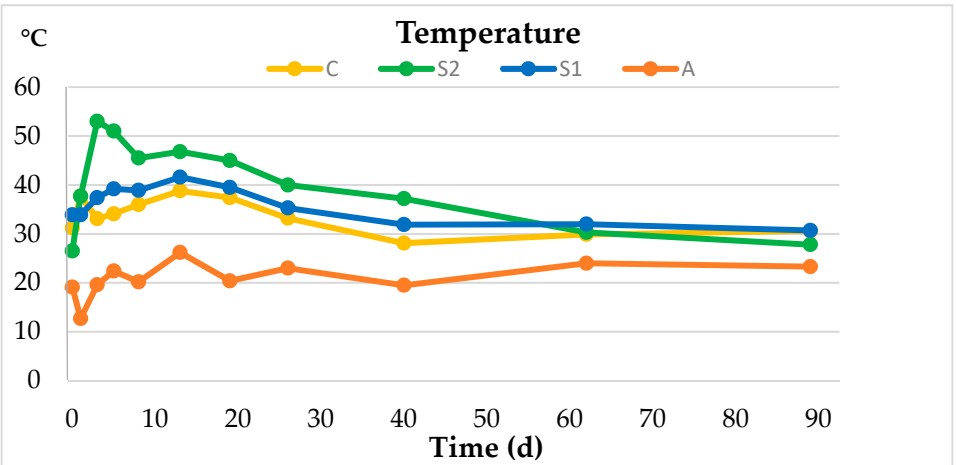

**Figure 1.** Changes in temperature in the core of the composted material. A—ambient temperature; C—control substrate; S1—manure amended with zeolite; S2—manure amended with a zeolite and lime.

In the initial stage of composting we observed a decrease in pH in the substrates without lime that could be attributed to formation of organic acids. Oxygen supply is important since the concentration of organic acids in composts becomes higher at low oxygen concentrations probably because of acid formation in anaerobic microenvironments [27]. According to Figure 2, from day 3 to 13 the pH in substrates S1 and C increased from 7.18 to 8.23 and from 7.35 to 7.91, respectively. In the S2 substrate, the initial abrupt increase was followed by decrease to 7.57 by day 3 and then increase to 8.58 by day 13.

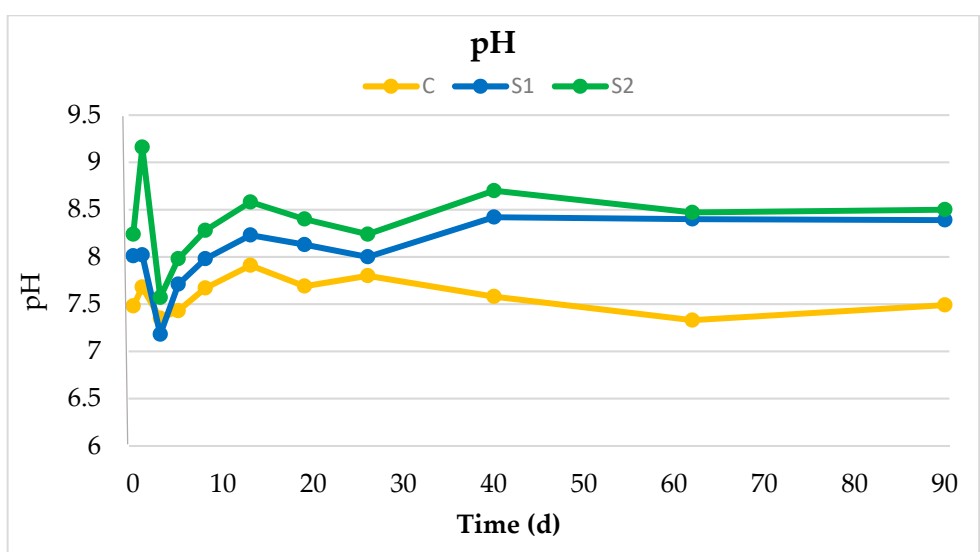

**Figure 2.** Evolution of pH during composting.

The increase after day 3 of storage could be attributed to biodegradation of the substrate and formation of ammonia. Then we can observe a decrease in S1 and S2 substrates to 8.0 and 8.24, respectively by day 26. After this period the pH levels changed to 8.42 and 8.7 in S1 and S2, respectively, and then remained constant until the end of observation. Throughout composting, pH in piles C, S1 and S2 was in the alkaline range and fluctuated from 7.18 to 9.16. While pH in C was significantly lower compared to S2 ($p > 0.05$), there was no significant difference in pH between C and S1. Significant difference in pH ($p < 0.001$) was detected between zeolite amended substrates S1 and S2. The pH level in the unamended control following day 26 showed a different course and reached pH 7.5 at the end of the experiment. In our zeolite-amended substrates S1 and S2, the pH of the compost at the end of the experiment was in the range of 8.39–8.5.

### 3.2. Dry Matter Content and Ash

The evolution of the dry matter content is the opposite of moisture and can be seen in Figure 3. This figure shows that the DM content increased from the initial 18.2%, 19.1% and 18.8% to the final 22.3%, 25.3% and 28.7% in C, S1 and S2, respectively. After day three of composting, it was significantly higher in S2 ($p < 0.001$) than in C and S1. No significant differences ($p > 0.05$) were detected between C and S1. The differences in the DM content are attributed to the action of the amendments. Higher dry matter values at the end of composting were reported also by other authors, however, their studies were conducted under different conditions and there are many factors that may affect the final results. There was no significant difference in moisture content between C and S1 ($p > 0.05$). Significant differences were detected between S2 than C ($p < 0.001$) and between S2 and S1 ($p < 0.001$).

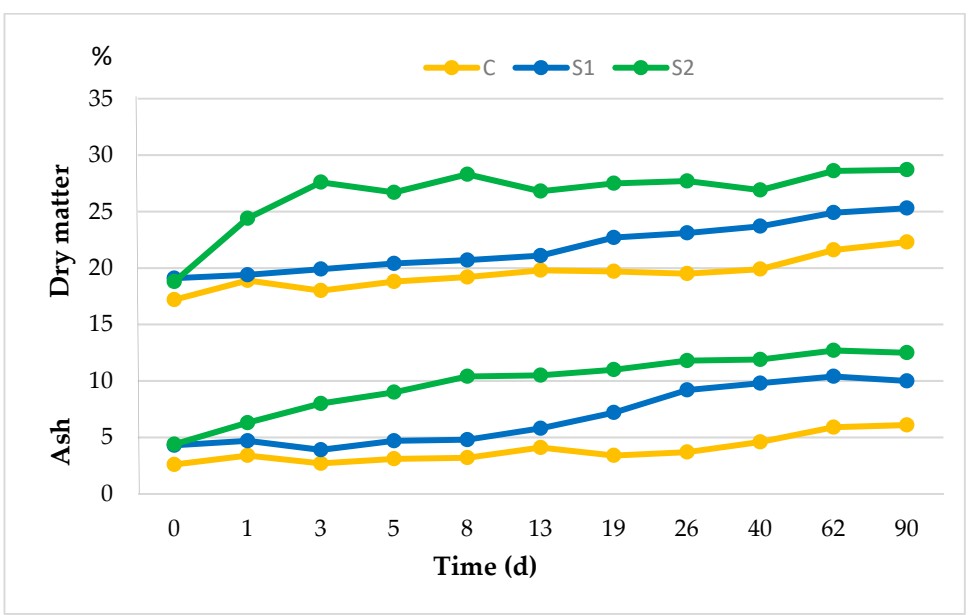

**Figure 3.** Variations in dry matter and ash during composting.

Ash represented the portion of substrates composed of a variety of inorganic minerals. The ash content at the start of composting was 2.6%, 4.3% and 4.4% for C, S1 and S2, respectively (Figure 3). Over time, it increased in relation with reduction of organic matter. Compared to the control, by day 26, the amount of ash in S1 and S2 increased up to 9.8% and 11.8%, respectively. Variance analysis showed significant difference ($p < 0.001$) between the S1 and S2 in the ash content. In substrate C ash was significantly lower than in S2 ($p < 0.001$), whereas no significant difference was detected between C and S1 ($p > 0.05$). The high accumulation of ash showed a high degree of degradation and volatilization. The reason for the high ash content in S1 and S2 may be the presence of zeolite. After 40 days

of composting, the ash content changed at much lower rate and at the end of observation reached 6.1%, 10.0%, 12.5% in C, S1, and S2 piles, respectively. The accumulation of ash is considered a good indicator of the decomposition processes.

### 3.3. C/N Ratio

Figure 4 shows a gradual decrease in the C/N ratio in all substrates during composting as a result of higher rate of C decomposition and lower N losses. The C/N ratio in C, S1 and S2 was 27.7, 29.2 and 32.0, respectively. In the C and S1 piles the C/N ratio decreased throughout the observation period. In the S2 substrate the C/N ratio increased to 34 during the initial two days and then dropped abruptly by day 5 to 29.0. The following decrease then more or less copied the trend followed by C and S1 substrates. After 40 days of observation the C/N ratio in all substrates dropped to the level of approximately 20. By the end of the observation period, the C/N ratio in C, S1 and S2 reached 17.0, 17.4, and 17.8, respectively, which indicated good maturation of the composted material. Variance analysis showed no significant differences ($p > 0.05$) in the C/N ratio in the substrates C, S1 and S2. In general, substrates containing cow manure used for composting have a low carbon to nitrogen ratio. The addition of a bulking agent helps to adjust the C/N ratios and bring them closer to the optimum values. According to the C/N ratios determined in our study, the substrates in C, S1 and S2 piles reached the mature state.

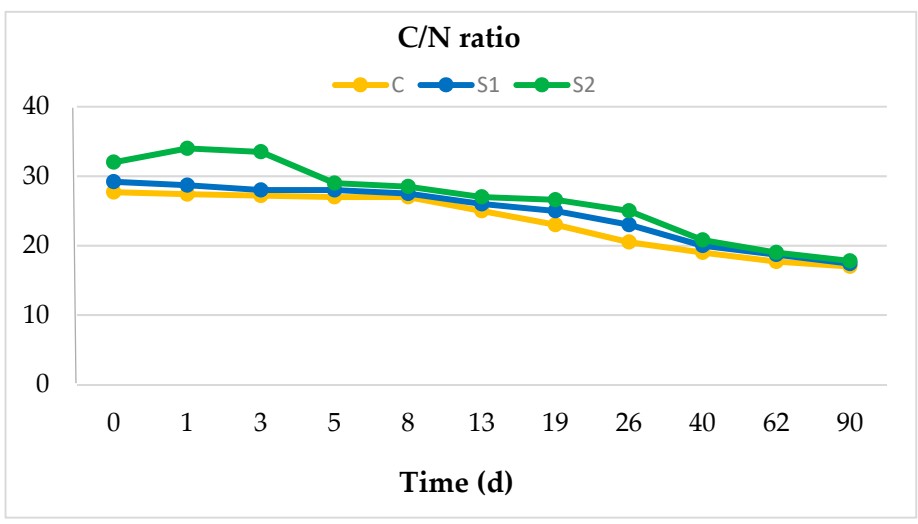

**Figure 4.** Changes in C/N ratio during composting.

### 3.4. Organic Matter Degradation

The initial OM content in the substrates C, S1 and S2 was 97.4%, 95.7% and 95.6%, respectively, and declined over time (Figure 5).

An analysis of the variance showed significant difference in organic matter loss among the substrates as follows: C and S1 ($p < 0.05$), C and S2 ($p < 0.001$) and S1 and S2 ($p < 0.01$). Generally, the gradual decrease in OM was caused by reduction of available carbon sources. The decrease in OM in the control substrate was the lowest. The difference between the initial and final content was only 3.5%. Lower OM degradation in C corresponded to lower pH and high moisture content which resulted in anaerobic conditions and lower rate of decomposition processes. As regards the S1, there was a significant decline between days 8 and 26 during which the OM dropped to 90.8%. Thereafter, there were no significant changes in the OM in this substrate until the end of observation. The maximum reduction was observed in S2 due to relatively rapid decomposition of organic matter by microorganisms that was facilitated by amendment with zeolite and lime. The OM content in this substrate decreased from the initial 95.6% to 89.6 by day 10 and to 88.2 by day 26. Reduction in OM is a good indicator of efficient degradation processes. After 90 days of composting, the final OM content was 93.9% in C, 90% in S1 and 87.5% in S2.

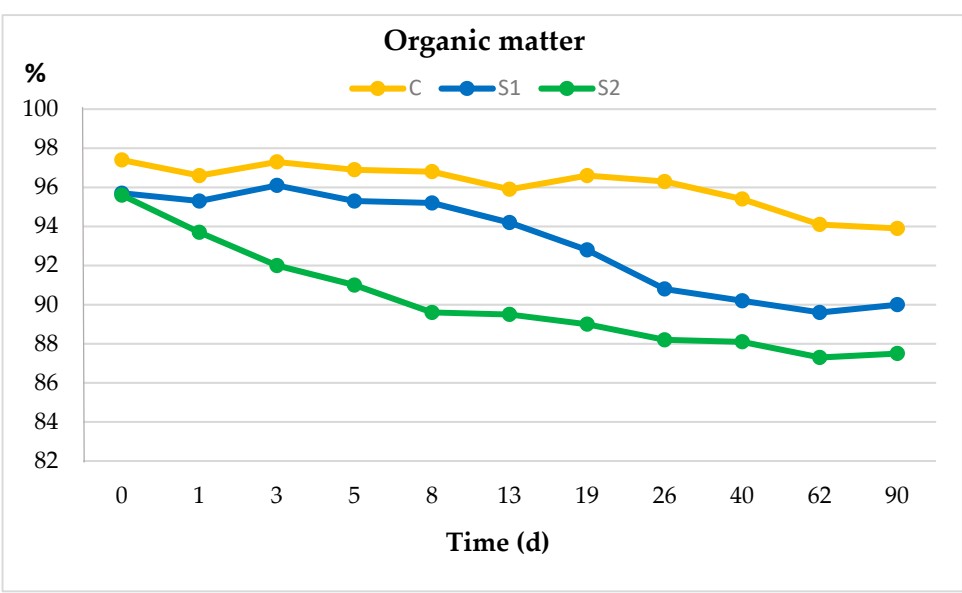

**Figure 5.** Organic matter degradation during composting.

### 3.5. Nitrogen Characteristics of the Compost

Figure 6 shows the level of N-NH$_4^+$ and N$_t$ in the investigated substrates. The initial levels of this parameter differed in individual substrates (6.29, 5.28 and 4.32 g·kg$^{-1}$, in S1, S2 and C, respectively). After day 26 the N-NH$_4^+$ concentrations gradually decreased in all substrates mostly due to conversion of N-NH$_4^+$ to volatile NH$_3$ and the immobilization of nitrogenous compounds (N-NH$_4^+$ was determined in 1:10 water extract). By day 90 of the observation there were very little differences between the substrates and the levels of N-NH$_4^+$ were very low: 0.32 g·kg$^{-1}$ in C; 0.20 g·kg$^{-1}$ in S1; 0.11 g·kg$^{-1}$ for S2. The final levels of N-NH$_4^+$ at the end of our study indicated that a mature compost was obtained. Variance analysis showed no significant differences in N-NH$_4^+$ concentration among S2, C and S1 ($p$ >0.05). Compared with C, the addition of zeolite and lime in S2 significantly reduced N-NH$_4^+$ release from this substrate ($p$ <0.05).

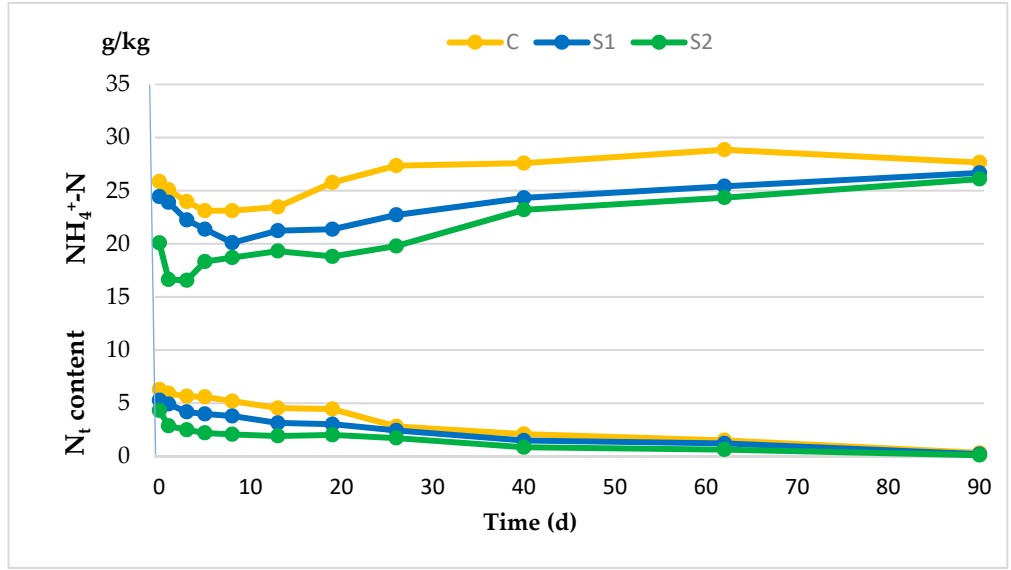

**Figure 6.** Changes in NH$_4^+$-N and N$_t$ content during composting.

The changes in N$_t$ over the time are illustrated in Figure 6. During the initial stage of composting (days 1 to 8), the N$_t$ content in substrates C and S1 decreased from 25.86

in C to 23.12 in C and from 24.44 to 20.11 in S1. The initial course of $N_t$ in pile S2 differed from those in the other two piles as $N_t$ in pile S2 dropped rapidly from 20.09 g·kg$^{-1}$ on day 0 to 16.57 g·kg$^{-1}$ on day 3, then started to increase but the increase was interrupted by a small drop by day 19. By day 26 all three substrates followed an increasing trend which continued always to the end in substrates S1 and S2 and only in the control $N_t$ decreased slightly over the last month of composting. Higher cumulative $N_t$ was reported for the C 28.86 g·kg$^{-1}$ in comparison with that in the S1 25.42 g·kg$^{-1}$ and S2 24.35 g·kg$^{-1}$. The final values of $N_t$ were 27.66 g·kg$^{-1}$, 26.68 g·kg$^{-1}$ and 26.1 g·kg$^{-1}$ for C, S1 and S2, respectively. Variance analysis showed significant differences in $N_t$ content of C and S2 ($p < 0.001$) and between S1 and S2 ($p < 0.05$). $N_t$ content was not significantly different between C and S1 ($p > 0.05$). The loss of $N_t$ at the beginning of the composting may be due to the loss of ammonia by volatilization at high temperatures.

*3.6. Bacteriological Examination*

Changes in the counts of total coliform bacteria and faecal coliforms are shown in Figure 7. The initial counts of coliform bacteria in C, S1 and S2 piles reached 7.591, 7.690 and 7.041 log$_{10}$ CFU·mL$^{-1}$, respectively. Throughout composting they decreased to 7.732, 6.531 and 5.279 log$_{10}$ CFU·mL$^{-1}$, respectively. The differences between C and S2 piles were significant at the level of $p < 0.001$ and between S1 and S2 at the level of $p < 0.01$ the differences between C and S1 were insignificant throughout composting ($p > 0.05$).

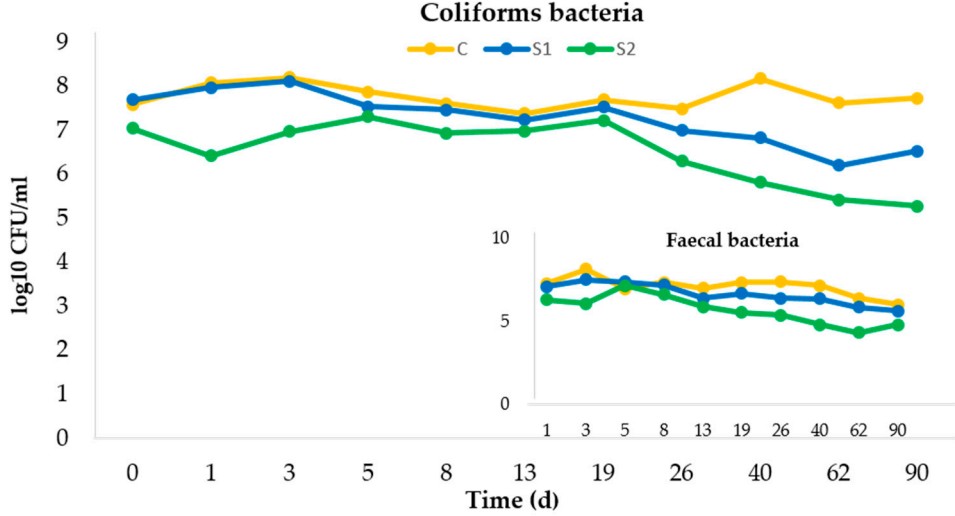

**Figure 7.** Counts of total coliforms and faecal coliform bacteria during composting.

The initial counts of faecal coliforms in C, S1 and S2 reached 7.398, 7.146 log$_{10}$ and 6.806 log$_{10}$ CFU·mL$^{-1}$, respectively. By the end of our investigation they declined to 5.991, 5.609 and 4.781 log$_{10}$ CFU·mL$^{-1}$, respectively. The differences in the counts of faecal coliform bacteria between C and S2 piles were significant at the level of $p < 0.001$ and those between S1 and S2 at the level of $p < 0.05$. The differences between C and S1 were insignificant ($p > 0.05$) throughout composting ($p > 0.05$).

The variations in the counts of faecal streptococci are shown in Figure 8. Their course was similar in all investigated piles throughout the composting. The initial counts of faecal streptococci were 7.869, 7.944 and 7.114 log$_{10}$ CFU·mL$^{-1}$ in C, S1 and S2, respectively, and by the end of composting they decreased to 4.602, 4.432 and 4.209 log$_{10}$ CFU·mL$^{-1}$, respectively. Their counts were slightly lower in S2 piles throughout composting compared to C and S1, the total amount of faecal streptococci in each substrate was between 4.209 log$_{10}$ CFU·mL$^{-1}$ and 7.944 log$_{10}$ CFU·mL$^{-1}$. Throughout composting there were no significant differences in the counts of faecal streptococci in C, S1 and S2 ($p > 0.05$).

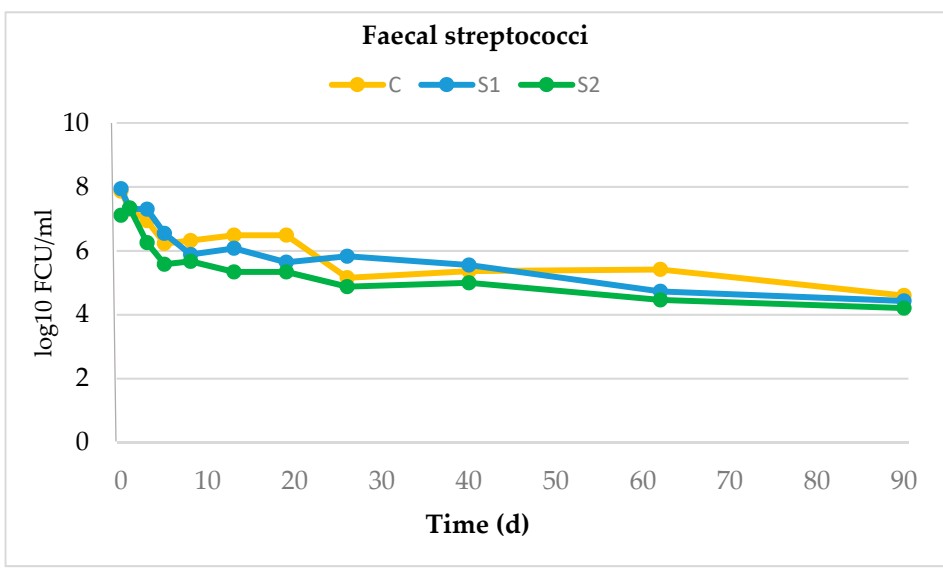

**Figure 8.** Counts of faecal streptococci during composting.

## 4. Discussion

The effects of additives on the composting process was investigated by many researchers. Their effect will differ, depending on their amount and properties, availability of oxygen, composition of substrate, moisture content, pH, C/N ratio and other factors [28].

Microbial activity during composting is mainly influenced by temperature, but other factors may also be important such as moisture, C/N, aeration and pH [28]. According to Antil et al. [29] the following temperature phases have been recognized in the composted substrate: (a) latent phase, which correlates with the time the microorganisms need to acclimatize and colonize the composted substrate; (b) growth phase, during which the biologically produced temperature rises to a mesophilic level; (c) thermophilic phase, when the temperature attains the highest level; (d) maturation phase, when the temperature decreases to mesophilic and at the end to the ambient level. The amplitude of temperature variation suggests a heterogeneity of the composting process, reflecting differences in the active microorganism populations.

In our study the temperature in thermophilic stage was lower in comparison with other studies which prevented effective reduction of potential pathogens. This may be due to the high initial moisture of all three substrates. The highest temperature was measured in the core of S2 pile where it raised almost to 55 °C, but for a short time only. The temperatures were higher also in S1 pile and exceeded those in the control. The course of temperatures corresponded with the DM content and indicated higher microbial activity in amended substrates. In the study by Meng et al. [7], that involved cow manure and corn straw composting, temperature in the piles increased spontaneously across both the mesophilic and thermophilic phases and was maintained above 55 °C for 30 days until it decreased during the cooling and maturation stages. Waste stabilization and pathogen destruction are most effective during the thermophilic phase [30]. Metabolic processes associated with biodegradation of compost result in a gradual decrease in metabolically derived heat to the mesophilic range [31].

According to Guo et al. [32], the specific physicochemical and biological properties of the composted materials affect the moisture content optimal for composting processes. High moisture leads to anaerobic conditions and results in slow temperature rise. According to Antil et al. [29] changes in water content varies with the waste to be composted, aeration and temperature and generally should be at 50–60%. If the compost is too wet, $O_2$ diffusion is inhibited and anaerobic conditions develop, which are undesirable because of the loss of N by denitrification, the rate of gas diffusion declines and the oxygen uptake rate becomes inadequate for metabolic demands of the microorganisms [9].

Amendment of manure with zeolite and lime in our study affected the dry matter content of the substrate. The DM increase in S2 and S1 piles could be attributed to the porous structure of zeolite and absorption of moisture in the microporous cavities. The water holding capacity of zeolite can reach up to 60% of its total weight. It is explained by high porosity of zeolites and their internal dehydrated cavities and channels [33].

The pH evolution observed in the present study was similar to that in the study by Huang et al. [34] who investigated changes in physicochemical characteristics during 70-day composting of swine, cattle and chicken manure without addition of bulking agents. The characteristics of the three investigated substrates differed. For cattle substrate the pH increased from 7.86 to 8.36 during the thermophilic phase, and then decreased sharply to 7.52 by the end of composting. Similar development was observed in the control substrate. After day 3, the pH increased over ten days from 7.86 to 8.36 in the thermophilic phase, and then decreased to 7.49 by the end of composting. The pH of the composted materials tends to decrease in the latter stage of composting due to nitrification that produces H+ [35]. The addition of zeolite and the related ammonium exchange affected pH of the substrate, so did the addition of lime. Adding lime can have adverse effect on microorganisms responsible for the composting process but, on the other hand, is beneficial when many acidic compounds are present in the pile. Our results are also in-line with those of Singh et al. [36], who observed that the presence of zeolite during composting caused slightly higher pH in comparison to unamended controls. Changes in pH caused by zeolite are attributed to its cation exchange and sorption abilities.

According to Villaseñor et al. [28] addition of zeolites can improve OM removal during composting by increasing porosity of the substrate. The lower OM degradation observed in our study corresponded to lower pH and higher moisture content. The higher moisture created anaerobic conditions what resulted in a poor composting process. Loss of organic matter reduces the weight of the pile, decreases the C/N ratio and reflects the efficiency of t degradation processes. Proper C/N ratio is important for determining the rate of decomposition of organic materials. In general, manures do not have the optimum C/N ratio. High C/N makes the composting process slow. Composting at lower initial C/N ratio can increase the loss of nitrogen as ammonia gas. Low C/N ratios can be corrected by adding a bulking agent to provide degradable organic-C [29]. Phosphogypsum and zeolite amendments decreased the carbon and dry matter losses in the presence of bulking agents. Combining sawdust as a bulking agent with amendments reduced the carbon and dry matter losses more efficiently than other combinations [35].

Evolution of nitrogen compounds is influenced by $N_t$, C/N ratio, degradable organic-C, particle size and the composting conditions, such as temperature and aeration. Emissions of nitrogen compounds can be released from compost as an elemental nitrogen ($N_2$) or nitrous oxide ($N_2O$) gases via nitrification and denitrification. The potential effect of $N_2O$ on global warming is about 300 times higher in comparison with $CO_2$. Up to 50% of the N in freshly excreted manure can be lost by conversion of urea to ammonia and the subsequent volatilization to the atmosphere. N losses during composting are associated mainly with the following three mechanisms: (a) volatilization of $NH_3$ at high temperatures and high pH values; (b) $NO_x$ volatilization attributed to nitrification and denitrification; (c) loss of water-soluble nitrogen due to leachate [17].

Our result and analysis of the variance showed significant differences in $N_t$ content between C and S2 ($p < 0.001$) and between S1 and S2 ($p < 0.05$). According to Antil et al. [29] pH is not a key factor for composting but is very relevant for controlling N-losses by ammonia volatilization, which can be particularly high at pH > 7.5. Loss of nitrogen through ammonia ($NH_3$) volatilization during composting of livestock manure causes nutrient imbalance in the compost [35]. The highest values of $NH_4^+$-N production occurred during the active phase of composting due to mineralisation of organic matter. Nitrification is limited during the thermophilic phase, because the high temperatures inhibit the action of the microorganisms responsible for the process. The emissions of $NH_3$ can cause nitrogen loss and acidification of compost and are one of the main malodorous compounds produced

during composting [37]. In our case, the addition of zeolite and lime to S2 significantly reduced N-NH$_4^+$ release from the composting substrate ($p < 0.05$) compared with the control. According to Yang et al. [17] the aerobic composting is an appropriate method for highly efficient nitrogen management during dairy manure composting. Substrate properties, mixing, temperature, pH and microbes of compost are interrelated and affect ammonia emissions. Modifying one parameter will affect the others. Nitrogen loss from composting mass, adsorption and precipitation of ammonia, could be improved by using zeolite [38].

According to Soudejani et al. [39], reduction of losses by leaching of nitrate from manure can be achieved by addition of zeolites. When N-NH$_4^+$ is available in the compost, clinoptilolite selectively absorbs N-NH$_4^+$ and makes it unavailable to the nitrifying bacteria. Using zeolites as additives in the fertilizers to control the retention and release of NH$_4^+$ reduces nitrogen losses, through cation exchange. Natural zeolites appear as cation exchangers because they have negative charge on the surface [21]. Many authors have reported that zeolites have potential to increase the rate of degradation, adsorption of NH$_4^+$ and reduction of NH$_3$ losses during composting [18,39].

Animal waste presents a source of disease causing microorganisms with zoonotic character, particularly representatives of the family *Enterobacteriaceae*, such as *Salmonella* sp. *Escherichia coli*, *Mycobacterium* sp., *Enterococcus* sp., *Streptococcus* sp., *Staphylococcus* sp. and other with threat to animals and man [40]. In our study the counts of coliform and faecal coliform bacteria showed significant differences between substrates C and S2 ($p < 0.001$). Acording to study of Meng et al. [7] NO$_3$-N, N-NH$_4^+$, C/N and temperature can significantly affect bacterial community succession only. On the other hand, total nitrogen and content of moisture significantly affected both, bacterial and fungal community. While total and ammoniacal nitrogen and pH significantly affected fungal abundance, total nitrogen and temperature had a significant effect on bacterial abundance. It is common belief that inactivation of pathogens during composting is attributed primarily to high temperatures (>55 °C) produced by the microbial activity. However, large surface-to-volume ratio of the mass may result in failure to achieve such lethal temperature levels as large portion of the produced heat is lost to the environment. Also microbial populations need proper proportions of carbon and nitrogen to grow and generate enough heat. In addition, low ambient temperatures may cause excessive loss of heat from the composted substrate to the environment [41].

## 5. Conclusions

Results of this study showed that addition of zeolite and hydrated lime to dairy manure containing straw as a bulking agent affected positively the decomposition processes during 3 months of stabilisation without turning. Higher temperature and dry matter content was detected in the amended substrates and evolution of pH was also positively affected. N-NH$_4^+$ content in the leachate of piles S1 and S2 was decreased which was attributed to selective absorption by zeolite. Reduction in the counts of investigated bacteria in the amended S1 and S2 piles may be associated with the influence of zeolite and lime, particularly with respect to temperature, pH and DM content. Results of this study indicate that the amendment of composted manure with inorganic materials such as zeolite and hydrated lime has a great potential in reducing nitrogen losses and contributing to improved stabilisation of substrates during composting.

**Author Contributions:** Conceptualization, E.Š. and N.S.; methodology, N.S.; software, E.Š.; validation, E.Š., N.S. and F.Z.; formal analysis, J.K. and I.M.; investigation, J.K. and I.M.; resources, M.V., K.V.L. and J.K.; data curation, E.Š., M.V., K.V.L., J.K. and N.S.; writing—original draft preparation, E.Š. and N.S.; writing—review and editing, N.S. and F.Z.; visualization, E.Š. and I.M.; supervision, N.S. and F.Z.; project administration, N.S. and F.Z.; funding acquisition, N.S. All authors have read and agreed to the published version of the manuscript.

**Funding:** This research was funded by Scientific Grant Agency of the Ministry of Education, Science, Research, and Sport of the Slovak Republic, projects number KEGA 001UVLF-04/2020, VEGA 1-

0529-19 and Visegrad Grant 22010056: Factors determining the occurrence of bovine mastitis in dairy farms situated in marginal regions.

**Conflicts of Interest:** The authors declare no conflict of interest. The funders had no role in the design of the study; in the collection, analyses, or interpretation of data; in the writing of the manuscript; or in the decision to publish the results.

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
