# Peer review of "Amendment of Livestock Manure with Natural Zeolite-Clinoptilolite and Its Effect on Decomposition Processes during Composting"

_agriculture, doi:10.3390/agriculture11100980_

Round 1
Reviewer 1 Report
A large number of scientific studies are devoted to various additives to manure in order to improve its properties. The relevance of this work lies in the fact that the authors propose to use such mineral additives that help to reduce the gases released into the atmosphere. Also, these additives increase the supply of nitrogen to the soil, increasing its fertility.
Some notes that would improve the manuscript:
1) lines 129-131. Are these the results of the authors? If not, then you need to add reference.
2) lines 146-147. You must specify the brand and manufacturer of hydrated lime.
Author Response
Dear reviewer, first of all, we would like to thank for assessing our paper and for valuable comments aimed at improvement of its quality. The comments will also help us when reporting our experiments in the future.
Lines 129-131:
Source of results: Central Agricultural Inspection and Testing Institute in Bratislava, Slovakia – inserted in the Reference list
Lines 146-147:
Brand and manufacturer of hydrated lime was inserted in the text.
Reviewer 2 Report
The paper of Subova et al. discuss the influence of zeolite and a mix of zeolite and CaO on the composting of cow manure. The paper presents the results of the determination of several parameters (temperature, pH, organic matter, total N, etc.) and attempts to provide an interpretation of the results. Overall this paper discusses and interesting topic and it is well suited for this journal. However, in my opinion several major changes need to be implemented . Some areas that need work are the following:
- Line 34 to 44. It would be nice to have some hard data. What is the amount of excrement produced? What does it mean “high” nutrient content? What are the type of pollutant for air, water and soil? Elaborate with quantitation
- Line 102-104. How is this reducing fertilizer? The addition of zeolites “dilutes” the nutrients so that overall more material will be necessary.
- Line 107. Hypothesis one. What are the physicochemical reasons by which lime would decrease the release of NH3? The most critical issue with this manuscript is that it does not seem to provide an interpretation of the data obtained with CaO with respect to those without CaO. Why the behavior of the zeolite with and without CaO differ in some cases? This needs to be addressed in detail
- Sections 2.2 is not clear. It seems that Table 1 refers to ranges provided by the manufacture? If the analysis of the actual samples was conducted the table should not report ranges but the actual number. Is 9wt% of cristobalite correct? This is a rare high temperature mineral which has little to do with the geological formation of zeolites in hydrothermal environments at much lower temperatures. Also it needs to be rectified what was the actual concentration of the clinoptilolite in the composting experiments? 2.5 wt% of the commercial material? So that would be 2.5wt% x 80% (according to table 1) so it is only 2% actually.
- Why where the lime and the zeolite mixed in equal weight % and not mol%? Line 23. How was the moisture determined? Line 28. What is partial exchange capacity? Line 141 à which temperature?
- Line 171. Explain the method for NH4 determination
- I think Figure 1 would be better if normalized as a percentage (Tpile/Tambient*100) or similar dimensionless parameter
- Line 219. With or without lime there seem to be a dip on day 3 so I do not think this has anything to do with very weak organic acids.
- Line 338. What is the evidence that NH3 gas escaped the pile of manure? Figure 8 and Figure N report the data based on kg of dry mass or wet mass? If wet mass was used all of this needs to be adjusted by moisture content. I may have miss it but I am not sure if the protocol for total N was given.
- Line 414. This needs a reference. The study was conducted at very high moisture level so how do the authors reconcile this with the rest of the literature? An attempt of explanation is provided in the following two paragraphs but it is not clear how the data can be normalized by moisture conditions or how any of the results can be translated to different moisture conditions. Also I do not understand how results on nitrogen can be compared if basically none of the other parameter I being kept constant. Each experiment is run at different temperature, different pH, and possibly different porosity. It is nice to collect data but what is their meaning? Overall the reader feels lost in this discussion. The authors seem to be wanting to describe and account for everything instead of addressing the hypothesis they had stated of understanding the effect of zeolite on the capture of ammonia
- Line 556. It is not clear why zeolite is what reduced NH3 and which data the authors have determined to univocally made such statements.
Formatting:
- The paper is particularly long and needs to be shortened substantially. The discussion is too long for the information presented and so is the results section. Several graphs can be pooled together as insets to each other or as a panel of graphs. For example dry ash could be an inset for dry matter. Same for ammonium nitrogen and total nitrogen and so on.
- “Materials and Methods” needs modifications. Sections 2.1 and 2.4 can be merged into one single paragraph. All the graphs seem to have been obtained from Microsoft Excel which makes them appearing neither appealing nor up to the international standard.
- The entire results section is purely descriptive, which is fine. However it needs to be shortened
- Line 526 to 537 can be removed
- The number of references should be reduced
Author Response
Dear reviewer, first of all, we would like to thank for assessing our paper and for valuable comments aimed at improvement of its quality. The comments will also help us when reporting our experiments in the future.
Lines 34-44
More detailed information (quantitation) about amount of produced animal excrements, their composition and pollutant released from manure to the environment would require many pages for the following reasons:
It is impossible to quantify reliably the amount of organic waste (livestock manure) produced annually on a worldwide scale as manure production differs according to animal species and age categories, varies widely by country, even by region and change constantly with time. Moreover, there are many other factors that must be considered, such as the system of housing (amount and quality of bedding, litterless system, way of manure handling and application, availability of runs or pasture), nutrition, level of hygiene, management practice and other. Even the reports about production of excrements by farm animals differ. For example, according to some reports, adult cow weighing 500 kg produces excrements (solids and urine) equal to 5-6 % its body weight but according to other sources up to 9 % b.w. The animal manure production estimated for the EU exceeds 1 400 million tons, the biggest contributor is France producing around 120 million tons per year not including manure deposited on pasture. Also the content of nutrients in manure will vary according to the system of keeping the animals mentioned above. Informative mean values for farmyard manure depending on the content of dry matter are as follows: N = 0.5 %, K = 0.59 %, P =0.14 %, Ca = 0.31 %. Poultry manure contains all 13 essential plant nutrients.
With respect to environmental pollution NH3 is one of the most frequently mentioned pollutants. Only the dairy and cattle manure accounts for above 40% of the total national NH3 emissions (50-85 % in China). Other important greenhouse gases produced during storage and handling of livestock manure are NH4 and N2O. Pollution of soil and water occurs when spreading manure beyond the capacity of the land or improper storage of farmyard manure allowing the dung water to reach the surrounding soil. Of biggest concern is the leaching of nitrates and phosphorus resulting in pollution of ground and surface water. Soil can be contaminated by copper (additive in animal feed), faecal microorganisms and antibiotics.
Lines 102-104:
Application of composted cattle manure without zeolite and manure amended with 7%, 14% and particularly of 21% of zeolite to sandy soil decreased chemical fertilizer application rate at two different irrigation regimes and improved quality and yield of sunflower. This was attributed to the lower rates particularly of nitrate leaching and less of P leaching. Evidently, retaining and slow releasing of nutrients due to the presence of zeolite was very important for the plants in this particular soil.
Line 107:
The acidic pH affects the rate of respiration of microbes and decreases the rate of degradation. The pH of the compost should be alkaline throughout and end of the composting process. Adding hydrated lime or other alkaline substances to raise the pH is one method to alleviate the inhibition caused by acids in compost and it functions well. However, it is not a perfect solution. First, liming involves extra costs; second, lime addition can be technically demanding; third, liming increases the ammonia emissions, thus increasing the environmental effects of ammonia release and reducing the nutrient status of the product. Adding zeolite together with lime decreases the volatilization of ammonia due to absorption of NH4+ in its porous structure.
Lime inhibits pathogens by controlling the environment required for bacterial growth. The high pH also provides a vector attraction barrier (i.e., prevents flies and other insects from infecting treated biological waste).
Lime treatment also reduces odours. It provides free calcium ions, which react and form complexes with odorous sulphur species such as hydrogen sulphide and organic mercaptans. Thus, the biological waste odours are not “covered over” with lime, but actually destroyed.
The benefit of adding lime to compost will continue through to the soil that the compost is applied to. Combined application of lime and organic manure remarkably increased the yield and nutrient uptake of the crops as well as improved the nutrient availability and other soil properties
Section 2.2 - changed to 2.1:
Information presented in Table 1 was obtained from the company Zeocem, a.s., Quarry in Nižný Hrabovec, Slovakia, which provided the zeolite clinoptilolite used in our study. They originated from the Central Agricultural Inspection and Testing Institute in Bratislava, Slovakia (2017). The actual concentration of clinoptilolite in composting experiments was aprox. 2.0 wt % but that of zeolite was 2.5 wt %, as reported.
To be able to use mol or mol %, elementary entity must be specified which is not possible for zeolite. Moisture content of manure was determined by drying samples of manure to a constant weight in an oven set to 105 oC.
Partial exchange capacity is the exchange capacity determined by using 0.15 M concentration of NH4Cl solution. In determination of total exchange capacity 5 M NH4Cl solution was used.
To obtain hydrated lime, limestone is first burned at temperatures of 1100°C or higher to drive out enough CO2 to leave just calcium oxide. – the temperature was added to line 141.
Line 171:
Determination of NH4+: Water extract (1:10) for ammonium determination was obtained by shaking homogenized samples for 15 minutes with distilled water. An appropriate aliquot of the sample was adjusted to pH 7.4 with a phosphate buffer and ammonia was distilled with steam to a receiving flask containing 0.02 N sulphuric acid. The excess of sulphuric acid was determined by titration with 0.02 N NaOH. After calculation of ammonium N (mg/l), the result was recalculated per DM.
Figure 1:
Temperature course is depicted in a way common for articles of similar orientation. We believe that the readers prefer to see the relevant temperature in the individual piles when checking the changes in other determined parameters.
Line 219:
During the initial phase of batch composting, reduced pH and high concentrations of organic acids can occur. During successful composting, the acids are decomposed and the pH increases. “Compared to well-aerated composting, composting at low oxygen levels results in a larger acid production and a slower break-down of acids (Beck-Friis et al., 2003)” ‒ information in the parentheses was inserted in the text.
However, composting is a complex process and the pH in composts is influenced by three acid-base systems. One is the carbonic system, with carbon dioxide, which is formed during decomposition and can escape as a gas or dissolve in the liquid, forming carbonic acid, bicarbonate and carbonate. This system tends to neutralise the pH of the compost, increasing low pH and reducing high pH. The second system is ammonium (NH4+) – ammonia (NH3), which is formed when protein is decomposed. During the initial phase of composting, most of the metabolised nitrogen is retained by growing microorganisms, but during the high-rate phase ammonia is released. The ammonia system increases the pH. The third system is composed of several organic acids, of which acetic and lactic acid dominate. This system can reduce pH down to 4.14 and it is important in the beginning of composting. These three systems combine to form the typical pH curve for batch composting, with initially falling pH and a sharp rise during high-rate degradation.
Line 338:
Volatilization of NH3 from manure at high pH is an accepted fact. However, how much NH4+ present in manure was converted to NH3 to the atmosphere depends on a number of factors, one of them the presence of zeolite.
Both figures (8 and 9) showing changes in nitrogen characteristics of the compost were constructed using data based on kg of dry mass.
Line 414:
With respect to moisture content during composting the following references are cited in the article: According to Guo et al. [32], the specific physicochemical and biological properties of the composted materials affect the moisture content optimal for composting processes. According to Antil et al. [33] changes in water content varies with the waste to be composted, aeration and temperature and generally should be at 50–60%.
Due to the porous structure of zeolite some moisture can be absorbed in its microporous cavities and affect the microenvironment. The water holding capacity of zeolite can reach up to 60% of its total weight. High initial moisture content was obviously one of the reasons why the temperatures in the core of the substrates did not reach high levels.
Line 560-567:
Conclusions were reformulated to reflect more accurately results of our study.
Formatting:
The results and discussion sections were shortened and several graphs were pooled as suggested by reviewer.
Sections 2.1, 2.2 and 2.3 in Material and Methods section were pooled.
Lines 530-541 were deleted.
The number of references was reduced.
Round 2
Reviewer 2 Report
The comments of the review were addressed by the authors.
Some graphs have improved and some parts of the manuscript improved as well. However, there were not highlighted changes in the revised manuscript which made difficult to understand which changes were actually made.
The first sentence in conclusion is missing the word "study" to read "results of this study showed"
Author Response
Responseses to reviewer’s comments
First of all, we would like to thank to both reviewers for assessing our paper and for their valuable comments aimed at improvement of its quality. The comments will also help us when reporting our experiments in the future.
Reviewer 1
Source of results: Central Agricultural Inspection and Testing Institute in Bratislava, Slovakia – inserted in the Reference list. Brand and manufacturer of hydrated lime was inserted in the text.
Reviewer 2
According the comments were formatted some changes in Introduction [detailed in manuscript attached].
Sections 2.1, 2.2 and 2.3 in Material and Methods section were pooled.
The results and discussion sections were shortened and several graphs were pooled as suggested by reviewer.
Lines 530-541 were deleted.
Conclusions were reformulated to reflect more accurately results of our study.
In the first sentence in conclusion was inserted the word "study" to read "results of this study showed".
The number of references was reduced.
